# Dietary Factors in Relation to Liver Fat Content: A Cross-sectional Study

**DOI:** 10.3390/nu12030825

**Published:** 2020-03-20

**Authors:** Cora Watzinger, Tobias Nonnenmacher, Mirja Grafetstätter, Solomon A. Sowah, Cornelia M. Ulrich, Hans-Ullrich Kauczor, Rudolf Kaaks, Ruth Schübel, Johanna Nattenmüller, Tilman Kühn

**Affiliations:** 1German Cancer Research Center (DKFZ), Division of Cancer Epidemiology, Im Neuenheimer Feld 581, 69120 Heidelberg, Germany; m.grafetstaetter@dkfz-heidelberg.de (M.G.); s.sowah@dkfz-heidelberg.de (S.A.S.); r.kaaks@Dkfz-Heidelberg.de (R.K.); ruth.schuebel@gmx.de (R.S.); 2Department of Diagnostic and Interventional Radiology, University Hospital Heidelberg, Im Neuenheimer Feld 110, 69120 Heidelberg, Germany; tno144@googlemail.com (T.N.); Hans-Ulrich.Kauczor@med.uni-heidelberg.de (H.-U.K.); johanna.nattenmueller@med.uni-heidelberg.de (J.N.); 3Medical Faculty, Heidelberg University, 69120 Heidelberg, Germany; 4Huntsman Cancer Institute and University of Utah, Department of Population Health Sciences, Salt Lake City, UT 84112-5550, USA; neli@hci.utah.edu

**Keywords:** non-alcoholic fatty liver disease, NAFLD, Mediterranean diet, DASH diet, food groups, MRI

## Abstract

Non-alcoholic fatty liver disease (NAFLD) can lead to functional liver impairment and severe comorbidities. Beyond energy balance, several dietary factors may increase NAFLD risk, but human studies are lacking. The aim of this cross-sectional study was to investigate the associations between food consumption (47 food groups, derived Mediterranean and Dietary Approaches to Stop Hypertension (DASH) diet quality scores) and liver fat content (continuous scale and NAFLD, i.e., >5% liver fat content). Liver fat content was measured by magnetic resonance imaging (MRI) in 136 individuals (BMI: 25–40 kg/m^2^, age: 35–65, 50.7% women) and food intake was recorded by food frequency questionnaires (FFQs). Associations between food items and liver fat were evaluated by multi-variable regression models. Intakes of cake and cookies as well legumes were inversely associated with liver fat content, while positive associations with intakes of high-fat dairy and cheese were observed. Only cake and cookie intake also showed an inverse association with NAFLD. This inverse association was unexpected, but not affected by adjustment for reporting bias. Both diet quality scores were inversely associated with liver fat content and NAFLD. Thus, as smaller previous intervention studies, our results suggest that higher diet quality is related to lower liver fat, but larger trials with iso-caloric interventions are needed to corroborate these findings.

## 1. Introduction

The global prevalence of non-alcoholic fatty liver disease (NAFLD) is rising, and NAFLD may affect approximately 25% of the adult world population [1]. NAFLD is characterized by a liver fat content of at least 5% [2] and used as an umbrella term for the different histological and clinical subtypes of a fatty liver [3]. NAFLD can progress into steatohepatitis [2], which may cause severe liver damage, leading to cirrhosis and hepatocellular carcinoma [4]. Since NAFLD is associated with metabolic risk factors, e.g., dyslipidemia, insulin resistance, hypertension, visceral obesity [2], it is considered as the hepatic manifestation of the metabolic syndrome [5].

The main goals in NAFLD management are moderate body weight reduction (7%–10%) and increased physical activity [6], while, despite first promising clinical studies with anti-diabetic and anti-inflammatory drugs [4], no specific medical treatment for NAFLD is approved so far [7]. Several studies have analyzed the relationships between dietary factors and NAFLD in recent years [7,8,9]. Epidemiological studies suggest that coffee consumption may decrease liver fat content [8], but little is known about associations between food consumption and NAFLD so far. As a dietary pattern, the Mediterranean (MED) diet has been recommended as a diet of choice for individuals with NAFLD [6], and observational studies [9], as well as smaller intervention studies [10,11], suggest potential benefits from following the MED diet. These effects could be related to dietary and nutrient composition (especially higher consumption of whole grains products, fruit, vegetables, extra virgin olive oil and fish, with an overall higher content of anti-oxidants) and go beyond those of calorie restriction and weight loss [12,13]. The Dietary Approaches to Stop Hypertension (DASH) diet that is also characterized by a higher proportion of fruits and vegetables may also lead to decreases in liver fat content according to a smaller intervention study [14].

Considering the limited evidence from human studies, it was the aim of this cross-sectional study to evaluate associations between food consumption and liver fat content. We used baseline data from the *Healthy Nutrition and Energy Restriction as Cancer Prevention Strategies: A Randomized Controlled Intervention Trial* (HELENA Trial) [15], a randomized controlled intervention study, in a post-hoc manner to evaluate food frequency questionnaire- (FFQ-) derived dietary items (47 food groups and two diet quality scores, the MED diet and DASH scores) in relation to liver fat content measured by magnetic resonance imaging (MRI) among 136 overweight or obese individuals.

## 2. Materials and Methods

### 2.1. Study Population and Design

Baseline data from 136 participants of the HELENA Trial, a randomized controlled intervention study carried out at the German Cancer Research Center (DKFZ), Heidelberg, Germany, were used in this study. The HELENA Trial (NCT02449148), which has been described in detail elsewhere [16], was designed to investigate the effects of intermittent calorie restriction (ICR) on metabolism and body weight compared to continuous calorie restriction (CCR). In total, 150 non-diabetic non-smokers (BMI: 25–40 kg/m^2^, age: 35–65, 50% women) were recruited between May 2015 and May 2016 and randomly assigned to either one of the three study arms: ICR, CCR, or control group. Exclusion criteria were prevalent diabetes mellitus (HbA1c ≥ 6.5% or fasting glucose > 126 mg/dL), renal dysfunction (increased creatinine, urea or uric acid), hepatic dysfunction (increased glutamic-pyruvic transaminase (GPT), GGT or GOT), cancer within the past 10 years, eating disorders, blood coagulation disorders, and impaired thyroid function. Further exclusion criteria were the use of certain medications (immunosuppressant or lipid-lowering drugs, diet pills, medication against menopausal symptoms), participation in intervention trials within the past three months, pregnancy or breastfeeding within the past twelve months, allergies (latex, disinfection, local anesthesia) and MRI contraindications (e.g., electronic implants, claustrophobia). Seven participants did not take part in the MRI assessment of the HELENA Trial, three individuals did not fill out the FFQ, and covariate information was missing for four participants; hence, we analyzed the associations between food consumption and liver fat content among 136 individuals. The HELENA Trial was approved by the ethics committee of the Heidelberg University Hospital (Heidelberg, Germany), and all participants provided written informed consent.

### 2.2. MRI Assessment

The MRI assessment was carried out by a 1.5 Tesla MRI scanner with a 70 cm bore design (MAGNETOM Aera; Siemens Healthcare, Erlangen, Germany) at the Department for Diagnostic and Interventional Radiology, University Hospital Heidelberg. A multi-echo GRE technique (Siemens LiverLab, Siemens Healthcare, Erlangen, Germany) was used [17]. MRI protocol, hardware and software were the same for all scans and one reader (TN) evaluated the imaging data on a post-processing software (OsiriX, Pixmeo SARL, Bernex, Switzerland). The proton density fat fraction (PDFF) was used to analyze three identical regions of interests (ROI, each area 4.00 cm²), dorsally, anterior-medially and anterior-laterally within the right liver lobe [17,18,19]. Results were validated by a second reader (JN), with intra- and inter-rater coefficients of correlation of 0.99 and 0.99.

### 2.3. Dietary Assessment and Diet Quality Scores

Food intake was recorded by a validated FFQ, which was developed at the German Institute of Human Nutrition (DifE), Potsdam, Germany [20]. Food groups were classified as proposed by Schulze et al. [21].

MED Diet Score

The MED diet score initially developed by Trichopoulou et al. is based on nine food items (vegetables, legumes, fruits, cereals, fish, olive oil, meat, dairy products, and alcohol) [22]. Since we had no data on olive oil consumption (and olive oil may be less frequently consumed in Germany than in Mediterranean regions), we used overall vegetable oil consumption instead of olive oil consumption for our MED diet score. All food groups beside alcohol were divided into sex-specific tertiles. For vegetables, legumes, fruits, cereals, fish and vegetable oil, two points were assigned for the highest tertile, one point for the middle tertile and zero points for the lowest tertile. The scoring for meat and dairy products was carried out assigning two points for the lowest tertile, one point for the middle tertile and zero points for the highest tertile. For moderate alcohol consumption (women: 5–25 g/day; men: 10–50 g/day), two points were given, while zero points were given for intakes outside of this range (see Appendix A
Table A1). The highest possible MED diet score was 18 points.

DASH Score

Eight food groups (fruits, vegetables, fiber, nuts and legumes, low-fat dairy products, sodium, meat and sweetened beverages) were taken into account for the calculation of the DASH score, as proposed by Fung et al. [23]. Intake levels were divided into sex-specific quintiles. The highest quintiles of fruits, vegetables, fiber, low-fat dairy products, nuts and legumes were assigned with five points; the lowest quintiles were assigned with one point. For sodium, meat and sweetened beverages, five points were given for the lowest quintile and one point was given for the highest quintile. Thus, 40 points were the theoretical maximum score (see Appendix A
Table A2).

### 2.4. Statistical Analyses

Statistical analyses were carried out using SAS Enterprise Guide 7.1 (SAS, Cary, NC, USA). Arithmetic means ± standard deviations (SD) were calculated for continuous variables and percentages for categorical variables for descriptive purposes. Associations between the consumption of individual food group items as well as diet scores and liver fat content in % were first analyzed by linear regression models, adjusted for calorie intake, sex, age, and waist circumference. In a second step, all food group items showing significant associations in the multi-variable model were further analyzed by backward elimination. Food groups still remaining significantly associated with liver fat content and diet scores were also evaluated in a logistic regression model in relation to the odds ratio of NAFLD (≥5% liver fat content).

## 3. Results

### 3.1. Characteristics of the Study Population

General characteristics of the study population are presented in Table 1. On average, the participants were 50.0 years old (individuals with NAFLD: 50.1 years, individuals without NAFLD: 49.8 years). Individuals in the NAFLD group had a higher education level, higher BMI values and higher blood levels of HbA1c, GPT, GOT, GGT as well as fasting glucose, fasting insulin and HOMA-IR. Men were more likely to be affected by NAFLD than women.

### 3.2. Food Consumption of the Study Population

Food consumption by food groups of the 136 HELENA participants is shown in Table 2. Individuals with NAFLD reported a higher consumption of meat, high-fat dairy products, and pasta and rice. Intake of fresh fruits, cakes and cookies, raw vegetables, and low-fat dairy products were higher among participants without NAFLD.

### 3.3. Diet Quality Scores

Diet quality scores are shown as the arithmetic means ± SDs (range) in Table 3. The overall mean MED diet score was 8.9 out of 18 possible points, with a slightly higher average value (9.5) among participants without NAFLD as compared to those with NAFLD (8.4). The average DASH score was 24.0 out of 40 points. Again, participants without NAFLD (24.9) showed slightly higher scores than individuals with NAFLD (23.2).

### 3.4. Associations between Food Items and Liver Fat Content

The food groups, which were significantly associated with liver fat content (%) in linear regression Model 1 are presented in Table 4. The consumption of cake and cookies as well as legumes showed inverse associations with liver fat content, while positive associations were observed for high-fat dairy and cheese consumption. All associations were modest in magnitude, with semi-partial R^2^ values ≤5% upon mutual adjustment for dietary items (Model 2).

We further carried out logistic regression analyses using the dietary variables that showed significant associations with liver fat content in linear regression models as independent variables. Odds ratios were obtained for the lowest quartile of intake compared to the highest quartile. The consumption of cake and cookies was inversely associated with NAFLD as a dichotomous outcome (i.e., liver fat content >5%, *p* = 0.02), with a 4.23-fold higher odds of NAFLD among participants in the lowest quartile of cake and cookie consumption (see Table 5). Given this unexpected strong positive association, and considering that FFQ-derived cake and cookie consumption may be affected by reporting bias [24], we further analyzed associations between cake and cookie consumption with FFQ-derived energy intake, estimated basal metabolic rate, and energy expenditure measured by accelerometer over one week [16] (see Appendix A
Table A3). Cake and cookie consumption was associated with higher reported energy intake, and with energy intake divided by the sum of estimated basal metabolic rate and activity energy expenditure, a measure of reporting bias [24]. However, when we adjusted linear and logistic regression analyses shown in Table 4 and Table 5 for the latter ratio, associations between cake and cookie consumption and liver fat content as well as NAFLD remained similar and statistically significant.

### 3.5. Associations between Diet Scores and NAFLD

#### Linear Regression Models

Results of the linear regression models on associations between diet quality scores (MED diet score, DASH score) and liver fat content (%) are shown in Table 6. Both scores showed significant inverse associations with liver fat content, although each score explained less than 4% of the variance in liver fat.

In logistic regression models, both diet quality scores showed significant inverse associations with NAFLD (see Table 7). The strengths of the associations between the two diet quality scores and liver fat as well as NAFLD were highly similar, and both scores were correlated (Spearman’s correlation of 0.59 for continuous diet quality scores).

### 3.6. Sensitivity Analyses

Further statistical adjustment of the linear and logistic regression models for circulating insulin, HbA1c and glutamic-pyruvic transaminase (GPT), which had been identified as correlates of NAFLD in the HELENA Trial in previous analyses [17], or for education level, physical activity and alcohol consumption only very marginally affected the observed associations. Adjusting the regression models on food groups and liver fat as well as NAFLD for the diet quality indices (MED diet score or DASH score) did not attenuate the observed associations.

## 4. Discussion

In the present study, we observed inverse associations between the intakes of cake and cookies as well as legumes with liver fat content in multi-variable linear regression models, while high-fat dairy and cheese consumption showed positive associations. The consumption of cake and cookies was also inversely associated with NAFLD as a dichotomous endpoint. Diet quality scores (MED diet score, DASH score) were significantly inversely associated with liver fat content and NAFLD.

To our knowledge, only one study concerning NAFLD and cake and cookie consumption has been conducted [25]. In this large cross-sectional analysis among 6671 participants of a population-based cohort from the Netherlands, cake consumption and hepatic triglyceride content were not associated in the multi-variable statistical model. The inverse associations of cake and cookie consumption with liver fat content and NAFLD observed in our study were unexpected. Due to a high sugar and fat content, cakes and cookies are rather considered as unhealthy, although epidemiological studies on cardiovascular diseases (CVDs) and diabetes have shown inverse associations with cake and cookie consumption [26,27,28,29]. In the EPIC-Europe Study, for example, participants who reported lower cake and cookie consumption had a higher risk of diabetes [26], which is in line with findings form cohorts in Sweden [27] and the USA [28]. Assuming that cake and cookies are mostly consumed in between meals, it was suggested that an improvement of glycemic control and insulin resistance through a constant carbohydrate supply is a potential biological mechanism underlying such associations [29]. However, it is also possible that differential misreporting of cake and cookie consumption leads to the inverse associations, as individuals with higher BMIs may report lower intakes of cakes and cookies [26]. In our study, there was no significant association between self-reported cake and cookie consumption and BMI, although we did observe a higher ratio of energy intake and energy expenditure (a marker of misreporting) among persons with higher cake and cookie consumption. Nevertheless, unlike in a previous study [24], statistical adjustment for misreporting did not affect the direction or magnitude of statistical associations in our study. Yet, we cannot rule out that the strong inverse association between cake and cookie consumption and NAFLD we observed was a chance finding.

Our analyses showed an inverse association between the consumption of legumes and liver fat content. Even though regular consumption of legumes may exert beneficial effects in the prevention of CVD and diabetes [30], legume consumption has not been analyzed in relation to NAFLD. However, reduced blood concentrations of cholesterol and triglycerides under (non-soy) legume-rich diets [31] as well as diets high in soy protein [32] have been described in meta-analyses, which is in line with findings from our study. Positive associations between high-fat dairy and high-fat cheese consumption have not been reported, and previous studies have mostly not shown associations between total dairy consumption and liver fat [25,33]. Considering that the present associations between liver fat and legumes, high-fat dairy, and high-fat cheese consumption were rather weak, and that no associations with NAFLD were observed, our findings should be interpreted with caution.

The MED diet score inversely associated with liver fat content and NAFLD in the present study. This result is in line with findings from other observational studies [34] and with experimental data to suggest protection from excess liver fat accumulation due to the high anti-oxidant contents of the MED diet [12,13]. Interestingly, in a recent cross-sectional study that reported an inverse association between adherence to the MED diet and NAFLD, the statistical effect was attenuated by adjusting for BMI [35], and iso-caloric dietary interventions trials are of special interest. In a small pilot intervention study, a significant decrease in liver fat with the MED diet was observed independently from weight loss [10]. Thus, results of the MEDINA Trial, a larger intervention study with the aim to verify the results of this pilot trial, are of special interest, but have not been published yet [36]. Adherence to the DASH diet was also associated with lower liver fat content in our trial. Hekmatdoost et al. showed in a case-control study that individuals in the highest DASH score quartile had a 30% lower risk of NAFLD; yet, when adjusting for BMI and dyslipidemia, the association was no longer significant. Razavi Zade et al. reported a beneficial effect of the DASH diet compared to a control diet over eight weeks on the metabolic status of individuals with NAFLD [14], although changes in liver fat content were not measured in this study, and further trials are needed to confirm potential effects of the DASH diet on liver fat content.

One limitation of our cross-sectional study is that due to the study design, no conclusions about temporal relationships between food consumption and liver fat content can be drawn. However, study participants probably did not know about their liver fat content during the study, and changes in dietary habits due to NAFLD seem unlikely, also given that BMI was not associated with cake and cookie consumption. As in any FFQ-based study, we cannot rule out misreporting of food consumption, which may have had an impact on the observed associations between cake and cookie consumption and liver fat in particular, although statistical adjustment for misreporting based on objectively measured energy expenditure had no effect on the present results. Still, as in any observational study, residual confounding cannot be excluded, and larger intervention trials are needed to overcome the limitations inherent to observational studies based on self-reported dietary data. The study sample was rather small and consisted of a homogenous group of metabolically healthy overweight people. Thus, there may have been a lack of contrast in dietary habits, and we may not have been able to detect weaker associations. At the same time, the present findings are consistent with previous findings from other study types and populations [12,13], and the fact that diet quality was inversely associated with NAFLD in this particular group of individuals does point to meaningful effects of diet on liver fat beyond energy balance. As stated above, however, larger iso-caloric interventions are needed to confirm the present observations. MRI is the most precise non-invasive method for NAFLD diagnosis [6] and therefore is a strength of our study compared to many studies using less accurate prediction scores based on blood markers for NAFLD assessment [17]. By design, smokers and individuals with impaired liver function were not included in the present study. Thus, self-reported alcohol consumption and circulating biomarkers of liver function were low and had no influence in statistical models, which indicates that confounding due to alcohol consumption did not affect our analyses.

## 5. Conclusions

The present study corroborates evidence to suggest that higher diet quality, as reflected by Mediterranean diet and DASH diet scores, is related to lower liver fat content. The possible associations between consumption of individual foods (cakes and cookies, legumes, high-fat dairy and high-fat cheese) with liver fat content requires replication in other studies, and iso-caloric intervention trials are needed to further evaluate effects of foods and complex dietary interventions on liver fat and NAFLD.

## Figures and Tables

**Table 1 nutrients-12-00825-t001:** Characteristics of the study population (*n* = 136).

	*Without NAFLD* *(Liver Fat Content <5%)*	*With NAFLD* *(Liver Fat Content ≥5%)*	*All*
n (%)	64 (47.1% ^1^)	72 (52.9% ^1^)	136
Sex n (%)			
*female*	37 (57.8% ^2^)	32 (44.4% ^2^)	69
*males*	27 (42.2% ^2^)	40 (55.6% ^2^)	67
Age	49.8 ± 8.3	50.1 ± 8.0	50.0 ± 8.1
University Degree *n* (%)	18 (28.1% ^2^)	29 (40.3% ^2^)	47
Height (cm)	170.7 ± 10.0	174.7 ± 9.1	172.8 ± 9.7
Weight (kg)	88.6 ± 12.7	99.5 ± 15.1	94.4 ± 15.0
Waist Circumference (cm)			
*females*	97.0 ± 9.8	102.9 ± 9.3	99.7 ± 9.9
*males*	102.5 ± 9.6	113.3 ± 9.4	108.9 ± 10.8
BMI (kg/m²)	30.4 ± 3.6	32.5 ± 3.6	31.5 ± 3.7
Calorie Intake (kJ/day)	10,206.7 ± 3396.7	10,380.6 ± 3288.1	10,298.8 ± 3328.3
Alcohol Intake (g/day)	10.4 ± 12.9	11.7 ± 14.4	10.9 ± 13.3
Liver Fat Content (%)	3.3 ± 0.9	11.2 ± 5.8	7.5 ± 5.8
HbA1c (%)	5.4 ± 0.3	5.5 ± 0.3	5.5 ± 0.3
GPT (U/L)	21.8 ± 9.3	30.8 ± 11.0	26.6 ±11.1
GOT (U/L)	21.3 ± 4.7	24.1 ± 5.3	22.8 ± 5.2
GGT (U/L)	22.7 ± 12.8	31.0 ± 17.8	27.1 ± 16.1
Fasting Insulin (mU/L)	404.6 ± 185.8	609.5 ± 301.6	513.1 ± 272.9
Fasting Glucose (mg/dL)	91.5 ± 6.8	95.6 ± 7.8	93.7 ± 7.6
HOMA-IR	2.2 ± 1.1	3.5 ± 1.7	2.9 ± 1.6

Values shown as the arithmetic mean ± SD; ^1^ row percentage; ^2^ column percentage. NAFLD, non-alcoholic fatty liver disease; GPT, glutamic-pyruvic transaminase.

**Table 2 nutrients-12-00825-t002:** Food consumption (gram/day) assessed by food frequency questionnaires among study participants with and without NAFLD.

	Without NAFLD, *n* = 64 (Liver Fat Content <5%)	With NAFLD, *n* = 72 (Liver Fat Content ≥5%)	All, *n* = 136
Whole grain bread	27.7 ± 23.7	26.8 ± 22.9	27.2 ± 23.1
Other bread	111.5 ± 82.2	105.0 ± 71.9	108.0 ± 76.7
Grain flakes, grains, muesli	8.8 ± 11.0	11.0 ± 13.7	9.9 ± 13.0
Pasta, rice	69.9 ± 48.7	71.4 ± 48.4	70.7 ± 48.4
Vegetarian dishes	13.7 ± 19.6	12.6 ± 12.4	13.1 ± 16.2
Chips	5.4 ± 6.3	7.0 ± 8.6	6.3 ± 7.6
Cake and cookies	60.9 ± 46.1	43.4 ± 31.3	51.6 ± 39.8
Confectionary	20.2 ± 19.0	21.0 ± 15.9	20.6 ± 17.4
Sweet bread spreads	13.8 ± 19.9	18.0 ± 23.7	16.0 ± 22.1
Eggs	12.2 ± 9.4	12.3 ± 9.8	12.3 ± 9.6
Fresh fruits	155.4 ± 121.0	126.0 ± 90.0	139.8 ± 106.3
Canned fruits	8.3 ± 24.5	5.4 ± 8.8	6.8 ± 18.0
Raw vegetables	84.4 ± 68.3	62.7 ± 47.1	72.9 ± 58.8
Cabbage	8.9 ± 9.9	10.7 ± 11.5	9.9 ± 10.8
Cooked vegetables	20.3 ± 17.0	20.8 ± 16.6	20.6 ± 16.7
Mushrooms	1.7 ± 1.9	1.8 ± 2.0	1.7 ± 1.9
Legumes	5.0 ± 4.9	5.0 ± 6.7	5.0 ± 5.9
Potatoes	39.4 ± 26.8	30.9 ± 22.3	34.9 ± 24.8
Fried potatoes	21.4 ± 22.0	22.2 ± 17.5	21.8 ± 19.7
Nuts	2.5 ± 2.7	3.8 ± 4.7	3.2 ± 4.0
Low-fat dairy products	106.5 ± 165.4	76.9 ± 96.7	90.8 ± 133.8
High-fat dairy products	46.6 ± 85.4	54.8 ± 81.0	50.1 ± 82.9
Low-fat cheese	4.0 ± 6.9	2.9 ± 9.4	3.4 ± 8.3
High-fat cheese	21.5 ± 20.1	23.3 ± 17.5	22.5 ± 18.7
Water	1080.9 ± 653.9	1081.4 ± 660.0	1081.2 ± 654.7
Coffee	451.2 ± 312.9	423.7 ± 312.9	436.7 ± 312.1
Decaffeinated coffee	13.8 ± 67.9	12.5 ± 67.1	13.1 ± 67.2
Tea	222.2 ± 330.1	234.5 ± 354.6	228.7 ± 342.1
Fruit juice	139.7 ± 237.4	126.4 ± 172.9	132.7 ± 206.1
Low-energy soft drinks	127.6 ± 418.0	105.7 ± 249.3	116.0 ± 338.1
High-energy soft drinks	78.5 ± 289.2	61.5 ± 110.9	69.5 ± 213.4
Beer	144.7 ± 272.6	289.8 ± 497.8	221.5 ± 412.6
Wine	133.3 ± 200.3	149.5 ± 231.3	142.9 ± 216.6
Spirits	1.1 ± 2.9	2.2 ± 4.3	1.7 ± 3.7
Other alcoholic beverages	39.8 ± 42.6	39.1 ± 54.9	39.4 ± 49.3
Butter	14.1 ± 15.5	13.4 ± 11.4	13.7 ± 13.4
Margarine	5.7 ± 9.9	6.1 ± 10.8	6.0 ± 10.4
Other vegetable fat	11.9 ± 9.4	9.6 ± 9.8	10.7 ± 9.6
Other fat	0.6 ± 1.2	0.7 ± 1.8	0.6 ± 1.5
Sauce	62.5 ± 36.9	61.8 ± 32.5	62.1 ± 34.5
Desserts	28.8 ± 34.1	32.9 ± 42.7	31.0 ± 38.8
Fish	15.7 ± 12.6	18.8 ± 15.3	17.3 ± 14.2
Poultry	20.6 ± 21.2	25.3 ± 23.9	23.1 ± 22.7
Meat	49.3 ± 39.5	55.1 ± 47.9	52.4 ± 44.1
Processed meat	35.8 ± 31.0	48.4 ± 37.3	42.5 ± 34.9
Soup	53.9 ± 41.7	59.3 ± 57.6	56.8 ± 50.6
Other	119.6 ± 58.8	111.2 ± 32.9	115.2 ± 46.9

Values shown as the arithmetic mean ± SD; the group ‘Other’ includes all food items which could not be assigned to one of the shown food groups.

**Table 3 nutrients-12-00825-t003:** Average diet quality scores of study participants with and without NAFLD.

	Without NAFLD, *n* = 64 (Liver Fat Content <5%)	With NAFLD, *n* = 72 (Liver Fat Content ≥5%)	All, *n* = 136
MED diet score	9.5 ± 3.1 (3–15)	8.4 ± 2.9 (2–14)	8.9 ± 3.0 (2–15)
DASH score	24.9 ± 4.9 (14–38)	23.2 ± 5.2 (12–36)	24.0 ± 5.1 (12–38)

Values shown as the arithmetic mean ± SD (range). MED, Mediterranean diet; DASH, Dietary Approaches to Stop Hypertension.

**Table 4 nutrients-12-00825-t004:** Associations between food consumption and liver fat content (%) (*n* = 136).

	*β*-Coefficient	Semi-Partial R^2^ (Type II)	*p*-Value
**Model 1**			
Cake and cookies	−0.20	3.7%	0.04
Legumes	−0.17	2.9%	0.02
High-fat dairy	0.25	5.7%	0.001
High-fat cheese	0.19	3.6%	0.01
**Model 2 (mutual adjustment for food groups)**			
Cake and cookies	−0.23	4.6%	0.002
Legumes	−0.15	2.0%	0.04
High-fat dairy	0.21	3.6%	0.007
High-fat cheese	0.17	2.6%	0.02
**Model 3 (additionally adjusted for the ratio of energy intake/total energy expenditure)**			
Cake and cookies	−0.23	4.6%	0.003
Legumes	−0.14	1.9%	0.049
High-fat dairy	0.21	3.5%	0.009
High-fat cheese	0.17	2.6%	0.02

Model 1: Linear regression adjusted for sex, age, waist circumference, calorie intake; Model 2: As Model 1, with further mutual adjustment for all dietary factors identified in Model 1; Model 3: As Model 2, with further adjustment for the ratio of energy intake/total energy expenditure.

**Table 5 nutrients-12-00825-t005:** Associations between food consumption and NAFLD (*n* = 136).

	Odds Ratio [95%-Confidence Interval] *	*P* for Trend –Value **
**Model 1**		
Cake and cookies	4.23 [1.32; 13.57]	0.02
Legumes	1.70 [0.56; 5.17]	0.81
High-fat dairy	0.34 [0.11; 1.05]	0.77
High-fat cheese	0.56 [0.18; 1.77]	0.48
**Model 2 (additionally adjusted for the ratio of energy intake/total energy expenditure)**		
Cake and cookies	4.39 [1.33; 14.51]	0.02
Legumes	1.78 [0.57; 5.55]	0.78
High-fat dairy	0.32 [0.10; 0.98]	0.76
High-fat cheese	0.55 [0.17; 1.76]	0.50

Model 1: Logistic regression adjusted for sex, age, waist circumference, calorie intake; Model 2: As Model 1, with further adjustment for the ratio of energy intake/total energy expenditure; * lowest quartile compared to highest quartile (Q1 vs. Q4); ** *p*-values were obtained for linear trend using dietary exposures as continuous predictors.

**Table 6 nutrients-12-00825-t006:** Associations between diet scores and liver fat content (%) (*n* = 136).

	*β*-Coefficient	Semi-Partial R^2^ (Type II)	*p*-Value
MED diet score	−0.195	3.44%	0.012
DASH score	−0.194	3.38%	0.013

Linear regression adjusted for sex, age, waist circumference, calorie intake.

**Table 7 nutrients-12-00825-t007:** Associations between diet quality scores and NAFLD (*n* = 136).

	Odds Ratio [95%-Confidence Interval] *	*P* for Trend –Value **
MED diet score	4.41 [1.28;15.15]	0.04
DASH score	4.41 [1.44;13.48]	0.05

Logistic regression adjusted for sex, age, waist circumference, and calorie intake; * lowest quartile compared to highest quartile (Q1 vs. Q4); ** *p*-values were obtained for linear trend using diet scores as continuous predictors.

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
