# Peer review of "Dietary Factors in Relation to Liver Fat Content: A Cross-sectional Study"

_nutrients, 2020, doi:10.3390/nu12030825_

Round 1
Reviewer 1 Report
- Introduction section: an healthy diet associated with physical activity could induce weight loss and control the progression of NAFLD. However, nutrients such as saturated fatty acids, trans-fats, simple sugars and animal proteins have a harmful effect on the liver. NAFLD patients usually follow Western diets which are rich in "junk food" and poor in cereals, whole grains, fruit, vegetables, extra virgin olive oil and fish. In this context, the Mediterranean diet is beneficial for NAFLD even when it is iso-caloric or there are no changes in body weight (PMID:31438482)
- Methods section: i suggest to include in the examined data HOMA-Ir to definethe degree of insulin resistance
- Discussion section: the Mediterranean diet is rich in food with antioxidants and polyphenols. Literature reports the ability of these bioactive molecules to influence the onset and development of cardio-vascular and metabolic diseases, including NAFLD (PMID:1438482)
Reviewer 2 Report
The aim of this cross-sectional study was to investigate the associations between food consumption (47 food groups, derived Mediterranean and DASH diet quality scores) and liver fat content (continuous scale and NAFLD, i.e. > 5% liver fat content). Associations between food items and liver fat were evaluated by multivariable regression models. The Authors concluded that higher diet quality is related to lower liver fat, but larger trials with iso-caloric interventions are needed to corroborate these findings.
My main concerns is about the relative small number of patients considered and on the potential bias deriving from self reporting the food intake.
Moreover the conclusion is well expected and not so novel.
Round 2
Reviewer 2 Report
The limits of the study remain but the authors well argue their thesis.
I have just one more small question. What is the unit of measurement in Table 2?
Author Response
We thank the reviewers for their very quick re-assessment of our revised paper. We have now added the unit to Table 2, as suggested by reviewer 2